# ICOS-deficient and ICOS YF mutant mice fail to control *Toxoplasma gondii* infection of the brain

Carleigh A. O'Brien¤, Tajie H. Harris*

Center for Brain Immunology and Glia, Department of Neuroscience, University of Virginia, Charlottesville, VA, United States of America

¤ Current address: Department of Psychiatry, Perelman School of Medicine, University of Pennsylvania, Philadelphia, PA, United States of America
* tajieharris@virginia.edu

**Data Availability Statement:** All relevant data are within the manuscript.

**Funding:** This work was funded by the National Institutes of Health grants R01NS091067 and R56NS106028 to T.H.H. and T32AI007496 to C.A.

## Abstract

Resistance to chronic *Toxoplasma gondii* infection requires ongoing recruitment of T cells to the brain. Thus, the factors that promote, sustain, and regulate the T cell response to the parasite in the brain are of great interest. The costimulatory molecule ICOS (inducible T cell costimulator) has been reported to act largely through the PI3K pathway in T cells, and can play pro-inflammatory or pro-regulatory roles depending on the inflammatory context and T cell type being studied. During infection with *T. gondii*, ICOS promotes early T cell responses, while in the chronic stage of infection ICOS plays a regulatory role by limiting T cell responses in the brain. We sought to characterize the role of ICOS signaling through PI3K during chronic infection using two models of ICOS deficiency: total ICOS knockout (KO) mice and ICOS YF mice that are unable to activate PI3K signaling. Overall, ICOS KO and ICOS YF mice had similar severe defects in parasite-specific IgG production and parasite control compared to WT mice. Additionally, we observed expanded effector T cell populations and a loss of Treg frequency in the brains of both ICOS KO and ICOS YF mice. When comparing the remaining Treg populations in infected mice, ICOS KO Tregs expressed WT levels of Foxp3 and CD25, while ICOS YF Tregs expressed significantly less Foxp3 and CD25 compared to both WT and ICOS KO mice. Together, these results suggest that PI3K-independent signaling downstream of ICOS plays an important role in Treg stability in the context of chronic inflammation.

## Introduction

Chronic infection with the eukaryotic parasite *Toxoplasma gondii* results in the recruitment of activated, Th1-polarized T cells to brain that contribute to control of the parasite and promote long-term host survival [1–4]. Though it has been well established that T cells play an essential role in the inflammatory immune response that restricts the replication and spread of the parasite in the brain, regulatory mechanisms that limit this inflammation are equally important in long-term host survival. Immunosuppressive cytokines such as IL-10 and IL-27 have been

O. and National Cancer Institute 5P30CA044579.
The funders had no role in study design, data
collection and analysis, decision to publish, or
preparation of the manuscript.

**Competing interests:** The authors have declared
that no competing interests exist.

shown to be required in the chronic phase of infection to limit excessive inflammatory cytokine production and T cell proliferation, both of which can lead to fatal immunopathology in mice lacking IL-10 or IL-27 signaling [5–7]. Regulatory T cells (Tregs) have also been implicated as an important player in limiting T cell responses during infection with *T. gondii*. During the acute stage of infection, increasing Treg numbers using treatment with IL-2 complexes resulted in decreased inflammation and immunopathology in the small intestine and liver, but was associated with increased parasite burden in the brain in the later stage of infection [8]. Furthermore, the recruitment and persistent presence of Tregs in the inflamed brain during the chronic stage of infection [9] suggests that this regulatory population could serve as an important means of limiting excessive inflammation in the CNS. The signals that serve to maintain the Treg population during the course of infection with *T. gondii*, however, are still being elucidated.

ICOS (inducible T cell costimulator) is one of a multitude of costimulatory molecules that has been shown to be important for optimal immune response to infections [10–13]. In particular, ICOS has been implicated in a wide array of immune functions, including the development of Tfh cells and formation of germinal centers that allows for optimal class-switched antibody production [14–17], as well as effector T cell proliferation and cytokine production [18–22]. ICOS does not only promote inflammatory T and B cell responses, however. More recently, ICOS has been shown to promote immune regulation rather than inflammation, either through supporting the survival and suppressive cytokine production of Tregs [23–27] or through promoting turnover of effector T cells through inhibiting IL-2 signaling [7]. These seemingly contradictory roles for a single costimulatory molecule emphasize the need to better understand the inflammatory context in which ICOS is playing a role, as well as how downstream signaling events following ICOS ligation are integrated in different cell T cell populations.

During the early acute stage of infection with *T. gondii*, ICOS KO mice have been reported to have sub-optimal T cell proliferation and IFN-γ production, though these mice were still able to control the parasite and survive to the later stages of infection [28]. In this case ICOS seems to play a redundant role with its family member CD28, as ICOS KO mice were only more susceptible to the infection on a CD28 KO background [28].We sought to further characterize the complex role of ICOS both at baseline and during chronic *T. gondii* infection in the CNS using two models of ICOS deficiency. We utilized a complete ICOS KO mouse, which lacks expression of ICOS on the surface of T cells and therefore lacks all signaling pathways downstream of ICOS, and ICOS Y$^{181}$F mice (hereby referred to as ICOS YF), which express normal levels of ICOS on the surface of T cells, but contain a tyrosine to phenylalanine mutation in the cytoplasmic tail of ICOS that prevents the recruitment and activation of PI3K [29]. We found that both ICOS KO and ICOS YF mice had baseline defects in maintaining Treg frequencies in the spleen, leading to skewed Teff:Treg ratios during homeostasis and in the absence of overt inflammation.

During chronic *T. gondii* infection, direct comparison of ICOS KO and ICOS YF mice to WT mice showed that both forms of ICOS deficiency resulted in severe defects in production of parasite-specific IgG, which correlated with higher parasite burdens in the brains of ICOS KO and ICOS YF mice compared to controls. Conversely, both ICOS KO and ICOS YF mice had expanded effector T cell populations in the spleen and brain during chronic infection compared to WT mice. This expansion of effector T cells correlated with a partial loss of Tregs in the spleens and brains of ICOS KO and ICOS YF mice. Interestingly, the remaining Tregs in the brains of ICOS KO and ICOS YF mice displayed distinct phenotypes with regards to several Treg identity markers. Though both genotypes showed a similar decrease in Treg frequency in the brain, the remaining ICOS KO Tregs still expressed WT levels of Foxp3 and

CD25. The remaining ICOS YF Tregs in the brain, on the other hand, had significantly decreased expression of both Foxp3 and CD25 compared to ICOS KO or WT Tregs. PI3K has long been assumed to be the major downstream signaling pathway activated with ICOS ligation, but these results suggest that PI3K-independent signaling pathways may be involved in shaping the Treg response during inflammation.

## Materials and methods

### Mice and infection model

C57BL/6 WT mice were purchased from Jackson laboratories to be used for age- and sex-matched controls. ICOS KO [30] and ICOS Y$^{181}$F (ICOS YF) [29] mice were kindly shared by Dr. Daniel Campbell from the Department of Immunology, University of Washington. ICOS KO and ICOS YF mice were then kept and bred in University of Virginia specific pathogen-free facilities, and were age- and sex-matched for experiments. All experimental procedures followed regulations set by the Institutional Animal Care and Use Committee at the University of Virginia. All infections used the type II parasite *Toxoplasma gondii* (strain Me49), which were maintained in chronically infected Swiss Webster mice (purchased from Charles River) and passaged through CBA/J mice (purchased from Jackson Laboratories) before experimental infections in C57BL/6, ICOS KO, and ICOS YF mice. For experimental infections, the brains of chronically infected (4 to 8 weeks) CBA/J mice were homogenized to isolate tissue cysts containing parasite. Experimental mice were then injected intraperitoneally with 10 to 20 parasite cysts.

### Tissue and blood processing

Chronically infected mice (5–6 weeks post infection) were sacrificed and perfused with 40 mL 1x PBS. Perfused brains and spleens were then put into cold complete RPMI (cRPMI) (10% fetal bovine serum, 1% NEAA, 1% pen/strep, 1% sodium pyruvate, 0.1% β-mercaptoethanol). Brains were then minced and passaged through an 18-gauge needle, and enzymatically digested using 0.227mg/mL collagenase/dispase and 50U/mL DNase (Roche) for 1 hour at 37˚C. Following enzymatic digestion, brain homogenate was passed through a 70 μm filter (Corning). To remove myelin and obtain a single cell suspension, filtered brain homogenate was resuspended in 40% percoll and spun at 650g for 25 minutes. Myelin was then aspirated and cells were washed with cRPMI and counted on a hemocytometer. Spleens were homogenized and filtered through a 40 μm filter (Corning). Red blood cells were then lysed using 2 mL RBC lysis buffer (0.16 M NH$_4$Cl). Following RBC lysis, spleen cells were then washed with cRPMI and resuspended for counting on a hemocytometer.

For experiments in which blood serum was collected, mice were sacrificed and the right atrium of the heart was cut to prepare for perfusion. Prior to perfusion, 300 μL of blood was taken from the chest cavity. To separate serum, blood samples were allowed to clot overnight at 4˚C. Blood samples were then spun at 14,000rpm for 10 minutes to fully separate clotted blood pellet from serum. Serum was stored at -80˚C until further analysis was performed.

### Flow cytometry

Following tissue processing, single cell suspensions from brains and spleens were plated in a 96-well U-bottom plate. Cells were first incubated with 50 μL Fc block (1μg/mL 2.4G2 Ab (BioXCell), 0.1% rat gamma globulin (Jackson Immunoresearch)) for 10 minutes at room temperature. Cells were then surface stained for CD3 (145-2C11), CD25 (PC61), CD8 (S3-6.7), CD4 (GK1.5), CD19 (eBio1D3), NK1.1 (PK136), MHCII (M5/114.15.2), CD11c (N418), CD80

(16-10A1), CD86 (GL1), CD45 (30-F11), CD11b (M1/70), and a live/dead stain for 30 minutes at 4˚C. After surface staining, cells were washed with FACS buffer (1% PBS, 0.2% BSA, and 2mM EDTA) and fixed at 4˚C overnight with a fixation/permeabilization kit (eBioscience) or 2% PFA. After overnight fixation, cells were permeabilized and stained for intracellular markers Bcl-2 (3F11) and Foxp3 (FJK-16S) for 30 minutes at 4˚C. Cells were then washed with FACS buffer, resuspended, and run on a Gallios flow cytometer (Beckman Coulter). All analysis was done using Flowjo software, v.10.

## ELISA

ELISAs for parasite-specific IgG were performed as previously described [31]. Briefly, Immunolon 4HBX ELISA plates (Thermofisher) were coated overnight at 4˚C with 5 μg soluble *Toxoplasma* antigen (STAg) diluted in 1X PBS. Following antigen coating, plates were washed with 1x PBS with 0.1% Triton and 0.05% Tween, then a blocking step was performed using 10% FBS for 2 hours at room temperature. After washing, serial dilutions of serum were added to plate wells overnight at 4˚C. Following incubation with serum samples, plates were washed as described above and wells were incubated with goat α-mouse IgG, human ads-HRP (Southern Biotechnology) for 1 hour at room temperature. Finally, ABTS peroxidase substrate solution (KBL) was added. Immediately after a color change occurred plates were read on an Epoch BioTek plate reader using Gen5 2.00 software.

## Parasite cyst counts

After mincing with a razor blade and before enzymatic digestion, brain tissue was passed through an 18- and 22-gauge needle. 30 μL of brain homogenate was then placed onto a microscope slide (VWR) and cysts were counted using a brightfield DM 2000 LED microscope (Leica).

## Statistical analysis

Statistical analysis comparing two or three different groups at a single time point was performed in Prism software (v. 7.0a) using a Student's t test or one-way ANOVA, respectively. When data from multiple experiments were combined, a randomized block ANOVA was used in R v.3.4.4 statistical software to account for natural variability between experimental dates by modeling the genotype (WT vs. ICOS KO vs. ICOS YF) as a fixed effect and the experimental date as a random effect. The particular test used for each individual panel shown, as well as the group size, is specified in each figure legend. P values are displayed as follows: ns = not significant, * $p < 0.05$, ** $p < 0.01$, *** $p < 0.001$. All data were graphed using Prism software and are presented as the mean ± standard deviation unless otherwise noted in the figure legend.

## Results

### ICOS KO and ICOS YF mice chronically infected with T. gondii have increased numbers of effector T cells in the brain as well as increased expression of CD25

The signals involved in the ongoing T cell response in the brain during chronic infection with *T. gondii* are still being elucidated. One such signal, ICOS, has previously been shown to play a costimulatory role early in the infection during T cell priming [18, 28]. In the chronic stage of infection, however, ICOS takes on a regulatory role by limiting effector T cells in the brain [7]. Furthermore, other models of inflammation have suggested additional roles for ICOS signaling in other areas of the immune response, namely the generation of Tfh cells [17, 32–34] and

the promotion of regulatory T cell activity [23, 25]. During a chronic infection that requires ongoing inflammation however, the seemingly complex and time-dependent role of ICOS remains to be fully described. In order to further elucidate the role of ICOS signaling during chronic infection, we utilized two genetic models of ICOS deficiency: total ICOS KO mice, which lack expression of ICOS on the cell surface and subsequently all downstream signaling pathways, and ICOS YF mice, which maintain wild-type levels of ICOS expression but contain a mutation in the cytoplasmic tail of ICOS that prevents activation of its major downstream signaling pathway PI3K [29, 35].

Given previous results suggesting that the loss of ICOS signaling exclusively in the chronic phase of infection leads to increased expansion of effector T cells in the inflamed brain [7], we hypothesized that genetic loss of ICOS expression of ICOS-mediated PI3K signaling would result in increased numbers of T cells in the brain during chronic infection. Indeed, both ICOS KO and ICOS YF mice had increased numbers of CD4$^+$ and CD8$^+$ effector T cells in the brain during chronic infection compared to WT mice (Fig 1A). Neither ICOS KO nor ICOS YF mice had a deficiency in expression of costimulatory molecules CD80 and CD86 on myeloid cells recruited to the brain during chronic infection (Fig 1B), suggesting that the increased accumulation of effector T cells in the brains of ICOS KO and ICOS YF mice was not a result of increased costimulation provided to T cells from antigen-presenting cells. Rather, the increased number of effector T cells correlated with increased expression of CD25 on the surface of effector CD4$^+$ and CD8$^+$ T cells in the brain (Fig 1C–1F), which is consistent with previously published results examining the effects of ICOS blockade only in the chronic phase of infection [7]. Increased expression of the survival factor Bcl-2 following antibody-mediated blockade of ICOS in the chronic phase of infection has previously been implicated as a mechanism supporting increased survival of effector T cell populations in the inflamed brain [7]. Interestingly, the increased number of effector T cells in the brains of ICOS KO and ICOS YF mice correlated with an increase in Bcl-2 expression in only the CD4$^+$ effector T cell population compared to WT mice, though no increase in Bcl-2 expression in the CD8$^+$ effector T cell population was observed (Fig 1G–1I), suggesting a differential effect of the genetic loss of ICOS or ICOS-mediated PI3K signaling on the CD4$^+$ and CD8$^+$ T cell populations in the inflamed brain. Taken together, these data suggest that a genetic loss of ICOS or ICOS-mediated PI3K signaling leads to the expansion of CD4$^+$ and CD8$^+$ effector T cells in the brain during chronic infection, supporting previously published results implicating an intrinsic inhibitory role for ICOS signaling on T cells in the inflamed brain.

The accumulation of effector T cells in the brains of ICOS KO and ICOS YF mice during chronic infection suggests a loss of regulation of the ongoing effector T cell response, could be due to an intrinsic effect of the loss of ICOS signaling in the effector T cells themselves as previously described [7]. However, the loss of regulation and subsequent expansion of effector T cell populations observed in ICOS KO and ICOS YF mice could also be explained by a lack of Treg-mediated suppression of T cell responses in the inflamed brain. ICOS has been shown to be important for Treg survival and maintenance of an activated, effector Treg population during both homeostasis and in models of inflammation [13, 23–27]. Therefore, we assessed the basic phenotype of the Treg population in the inflamed brain during chronic *T. gondii* infection in ICOS KO and ICOS YF mice. We observed a similar decrease in the frequency of Tregs in the brains of ICOS KO and ICOS YF mice, as both genotypes presented with about half the frequency of Tregs in the brain as WT controls (Fig 2A–2D), though total Treg numbers were not decreased (Fig 2E). Interestingly, however, of the Tregs that were found in the brain during chronic *T. gondii* infection, WT and ICOS KO Tregs maintained equivalent levels of Foxp3 expression, yet ICOS YF Tregs in the brain had significantly decreased levels of Foxp3 expression compared to both WT and ICOS KO Tregs (Fig 2F). Foxp3 expression has been

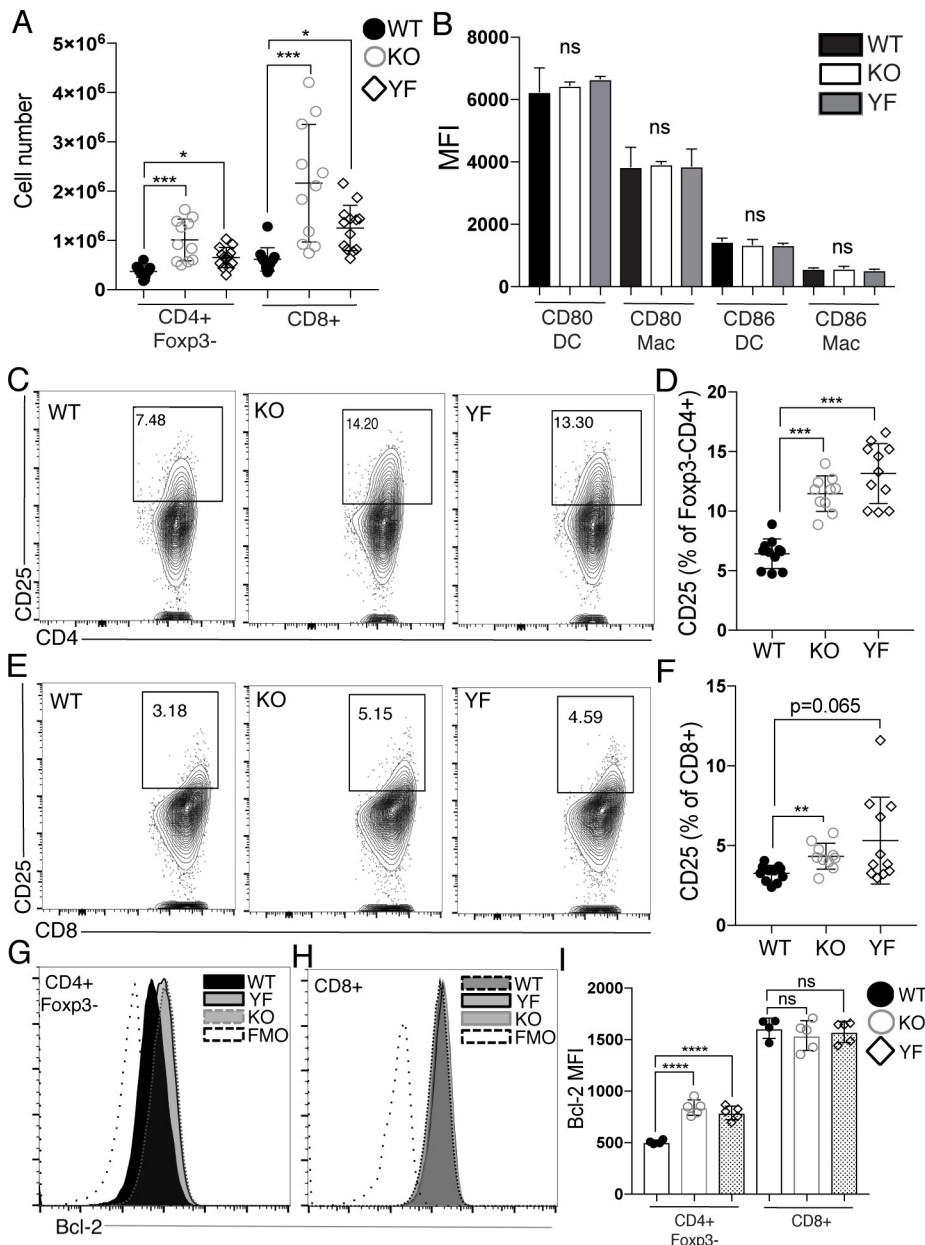

**Fig 1. ICOS KO and ICOS YF mice chronically infected with *T. gondii* have increased numbers of effector T cells in the brain as well as increased expression of CD25.** WT, ICOS KO, and ICOS YF mice were infected with *T. gondii*. 5 weeks post-infection, brain mononuclear cells (BMNC) were isolated and analyzed by flow cytometry. **(A)** Total numbers of effector CD4+ T cells (CD4+Foxp3-) and effector CD8+ T cells isolated from the brain (n = 3–5 per group, data is pooled from three independent experiments and analyzed by randomized block ANOVA). **(B)** Mean fluorescence intensity (MFI) of CD80 and CD86 on macrophages and DCs isolated from the brain. DCs were pre-gated on live CD45+CD19-NK1.1-CD3-MHCIIhiCD11c+ and macrophages (Mac) were pre-gated on CD45+CD19-NK1.1-CD3-CD11b+MHCIIhi/low (n = 4–5 per group, data is representative of two independent experiments and analyzed by randomized block ANOVA). Representative flow plots and quantification of CD25+ CD4+ effector T cells **(C-D)** and CD8+ effector T cells **(E-F)** (n = 3–5 per group, data is pooled from three independent experiments and analyzed by randomized block ANOVA). Representative histograms of Bcl-2 expression of CD4+ effector T cells **(G)** and CD8+ effector T cells **(H)**. **(I)** MFI of Bcl-2 on effector CD4+ and CD8+ T cells in the brain (n = 4–5 per group, data is representative of three independent experiments and analyzed by one-way ANOVA). ns denotes not significant, * denotes p<0.05, ** denotes p<0.01, *** denotes p<0.001 and **** denotes p<0.0001 for all panels.

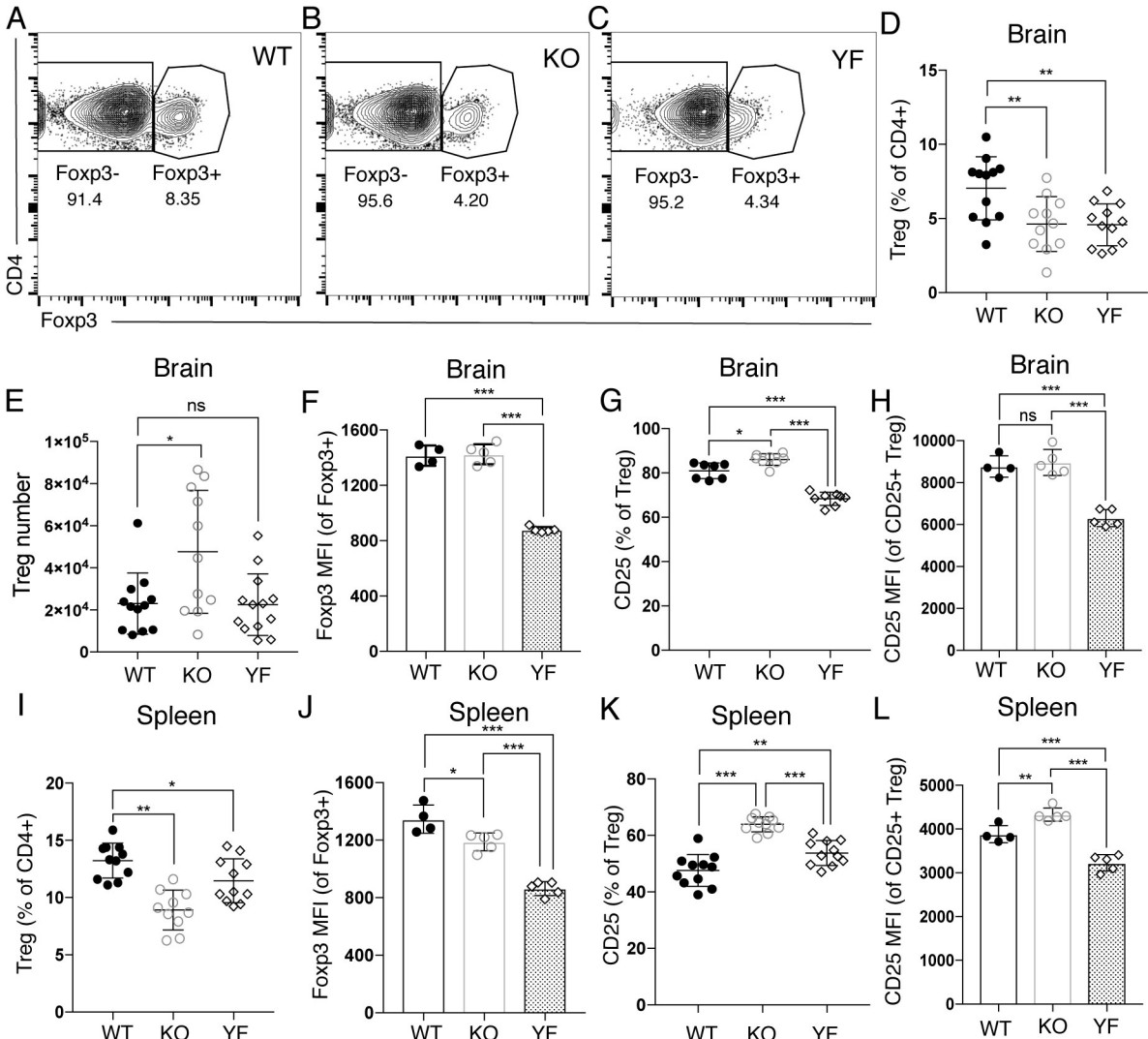

**Fig 2. ICOS KO and ICOS YF mice have distinct Treg defects in the brain during chronic *T. gondii* infection.** Immune cells were isolated from the brains **(A-H)** and spleens **(I-L)** of chronically infected WT, ICOS KO, and ICOS YF mice and analyzed by flow cytometry. **(A-C)** Representative flow cytometry plots are shown indicating the decrease in Treg frequency in the brain observed in ICOS KO and ICOS YF mice during chronic infection. **(D)** Total Treg frequency in the brain (n = 3–5 per group, data is pooled from three independent experiments and analyzed by randomized block ANOVA). **(E)** Treg number in the brains of WT, ICOS KO, and ICOS YF mice (n = 3–5 per group, data is pooled from three independent experiments and analyzed by randomized block ANOVA). **(F)** MFI of Foxp3 in the Treg population isolated from the brain during chronic infection (n = 4–5 per group, data is representative of three independent experiments and analyzed by one-way ANOVA). **(G)** The frequency of CD25+ Tregs in the brain (n = 3–5 per group, data is pooled from three independent experiments and analyzed by randomized block ANOVA). **(H)** MFI of CD25 on CD25+ Tregs in the brain (n = 4–5 per group, data is representative of three independent experiments and analyzed by one-way ANOVA). **(I)** The frequency of Tregs among the CD4 + population in the spleen during chronic infection (n = 3–5 per group, data is pooled from three independent experiments and analyzed using randomized block ANOVA). **(J)** The MFI of Foxp3 on Tregs in the spleen during chronic infection (n = 4–5 per group, data is representative of three independent experiments and analyzed using one-way ANOVA). **(K)** The frequency of CD25+ Tregs in the spleen during chronic infection (n = 3–5 per group, data is pooled from three independent experiments and analyzed by randomized block ANOVA). **(L)** MFI of CD25 on CD25+ Tregs in the brain (n = 4–5 per group, data is representative of three independent experiments and analyzed by one-way ANOVA). * denotes p<0.05, ** denotes p<0.01, and *** denotes p<0.001 for all panels.

previously shown to regulate expression of the high affinity IL-2Rα chain (CD25) [36, 37], and both Foxp3 and CD25 are required for the development and maintenance of a suppressive regulatory T cell population [38–41]. Given the decreased expression of Foxp3 in ICOS YF Tregs, we also assessed the expression of CD25 on the Treg population in the inflamed brain during

chronic infection. Similar to the decreased expression of Foxp3, we observed significantly lower levels of CD25 expression on ICOS YF Tregs in the brain compared to Tregs from WT or ICOS KO mice (Fig 2G and 2H).

To determine if the decrease in Foxp3 and CD25 expression was specifically found on ICOS YF Tregs in the inflamed brain, we also assessed the phenotype of Tregs found in the spleen at the same time point during chronic infection. Similar to the decreased Treg frequency observed in the chronically infected brain, we found a decreased frequency of Tregs in the spleens of both ICOS KO and ICOS YF mice during the same stage of chronic infection (Fig 2I). Unlike in the brain, however, the expression of Foxp3 on the remaining Tregs in the spleen was significantly decreased in both ICOS KO and ICOS YF mice compared to WT controls, though Foxp3 expression in ICOS YF Tregs in the spleen was decreased to a greater degree than that seen in ICOS KO Tregs (Fig 2J). Additionally, in contrast to the ICOS YF-specific defect in CD25 expression on Tregs found in the inflamed brain during chronic infection, both ICOS KO and ICOS YF Tregs in the spleen had an increased proportion of Tregs expressing CD25 (Fig 2K), though only ICOS KO Tregs in the spleen were expressing more CD25 than WT Tregs on a per-cell basis (Fig 2L). Overall, a partial loss of Treg frequency was observed in both ICOS KO and ICOS YF mice in both the brain and spleen during chronic infection. Interestingly though, the dynamics of Foxp3 and CD25 expression in ICOS KO and ICOS YF Tregs is distinct in different tissues. While ICOS YF Tregs had defects in both Foxp3 and CD25 expression on a per-cell basis in both the brain and the spleen, ICOS KO Tregs only had defects in Foxp3 expression in the spleen. These results suggest that, while ICOS expression is required for the maintenance of the Treg population in the inflamed brain and spleen, the ICOS YF mutation specifically leads to differences in retaining canonical Treg markers such as Foxp3 and CD25 in the brain during chronic infection.

Not only does ICOS play a role in effector and regulatory T cell responses, but ICOS signaling also plays a role in the development of Tfh cells, the formation of germinal centers, and the optimal production of class-switched antibody [14, 15, 17, 32, 42, 43]. Based on these previous reports, it was possible that ICOS KO and ICOS YF mice fail to produce adequate parasite-specific antibody during infection, therefore leading to increased parasite burdens in the brains of chronically infected animals. To address this question, we first looked for parasite-specific IgG levels in the serum of chronically infected WT, ICOS KO, and ICOS YF mice. Similar to what has been previously reported regarding antibody responses in ICOS KO and ICOS YF mice [29], both genotypes showed extremely diminished levels of circulating parasite-specific IgG in the serum during chronic infection compared to WT mice (Fig 3A). μMT mice that lack B cells succumb to *T. gondii* infection in the chronic stage with high parasite burdens found in the brain, suggesting that antibody responses are crucial in limiting parasite and promoting the long-term survival of mice infected with *T. gondii* [44]. We hypothesized that ICOS KO and ICOS YF mice, because of their inability to produce parasite-specific IgG, would also have defects in parasite control and increased parasite burdens in the brain during chronic infection. Indeed, both ICOS KO and ICOS YF mice had significantly increased parasite cyst burdens in the brain compared to WT controls (Fig 3B). This increase in cyst number was further shown in H&E stained histology sections from the brains of chronically infected WT, ICOS KO, and ICOS YF mice. Interestingly, while chronically infected WT mice usually have single parasite cysts in isolation from others, both ICOS KO and ICOS YF mice had and increased incidence of parasite cyst "clusters", where multiple cysts were found in close proximity to each other in the inflamed brain (Fig 3C). Taken together, these results suggest that ICOS KO and ICOS YF mice cannot produce sufficient parasite-specific class-switched antibody to control parasite, which is likely a contributing factor to the observed increase in parasite burden in the brains of chronically infected ICOS KO and ICOS YF mice compared to controls.

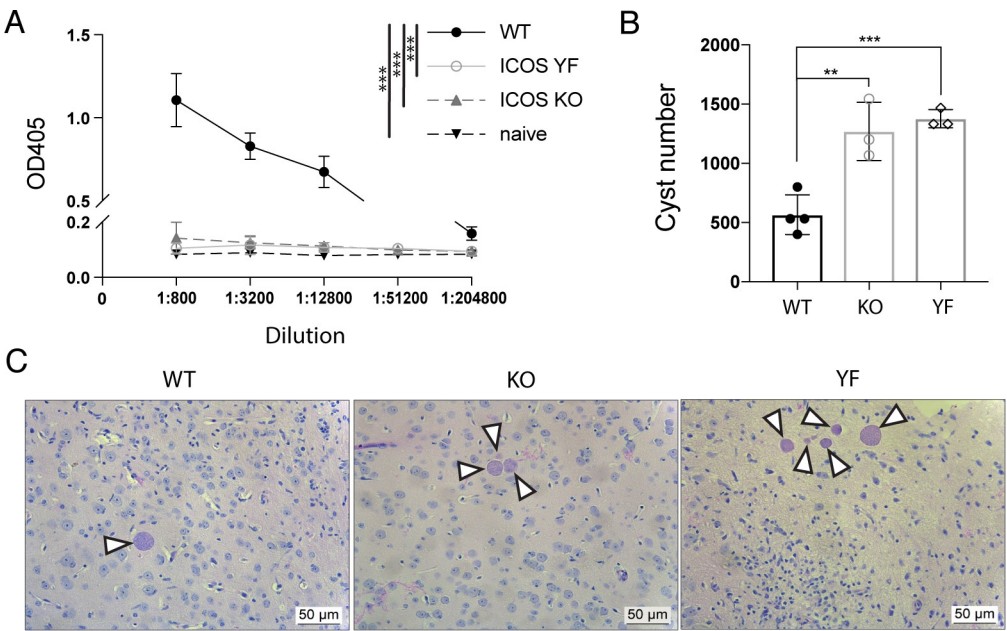

**Fig 3. ICOS KO and ICOS YF mice have severely impaired parasite-specific IgG antibody production and have increased cyst burden in the brain compared to WT infected mice. (A)** Parasite-specific total IgG was measured by ELISA using serum from chronically infected WT, ICOS KO, and ICOS YF mice as well as naïve controls (n = 3–4 per group, data is representative of two independent experiments and analyzed using two-way ANOVA). **(B)** Total cyst numbers from the brains of chronically infected mice were enumerated by light microscopy (n = 3–4 per group, data is representative of two independent experiments and analyzed by one-way ANOVA). **(C)** Representative H&E stained sections from the brains of chronically infected WT, ICOS KO, and ICOS YF mice. White arrows indicated individual parasite cysts ranging in size. Scale bar is 50 μm. ** denotes p<0.01, and *** denotes p<0.001 for all panels.

Because ICOS KO and ICOS YF mice lack ICOS or ICOS-mediated PI3K signaling, respectively, from birth, it was possible that some of the reported T cell abnormalities, namely the expanded effector T cell populations and Treg defects, might be observed at baseline before the onset of inflammation. To address this, we analyzed the T cell populations in the spleen during homeostasis in adult WT, ICOS KO, and ICOS YF mice. Importantly, the baseline studies shown using ICOS KO and ICOS YF mice were performed on different experimental days, so direct comparison between ICOS KO and ICOS YF mice should not be made in this case. At baseline, ICOS KO and ICOS YF mice have a decreased frequency of Tregs in the spleen compared to their respective age-matched WT controls (Fig 4A and Fig 4C). Interestingly, of the Tregs isolated from the spleen, both ICOS KO and ICOS YF Tregs showed decreased levels of Foxp3 (Fig 4B and Fig 4D). These results are reminiscent of the decreased Foxp3 expression observed in ICOS KO and ICOS YF Tregs in the spleen during chronic infection (Fig 2H), though ICOS KO Tregs found in the inflamed brain expressed normal levels of Foxp3, while ICOS YF Tregs did not (Fig 2E). Taken together, these results suggest that the defects in Foxp3 expression in ICOS KO and ICOS YF Tregs in the spleen during chronic infection could stem from a baseline defect in Foxp3 expression present before any active inflammation.

To determine whether the Treg defects observed in ICOS KO and ICOS YF Tregs at baseline had any effects on the suppression of effector T cell populations, we enumerated effector T cells in the spleen as well. Interestingly, we observed significantly expanded effector T cell populations in the spleens of both ICOS KO and ICOS YF mice compared to their respective WT controls (Fig 4E and 4F). A skewed CD4$^+$ effector T cell: Treg (Teff/Treg) ratio in the spleen

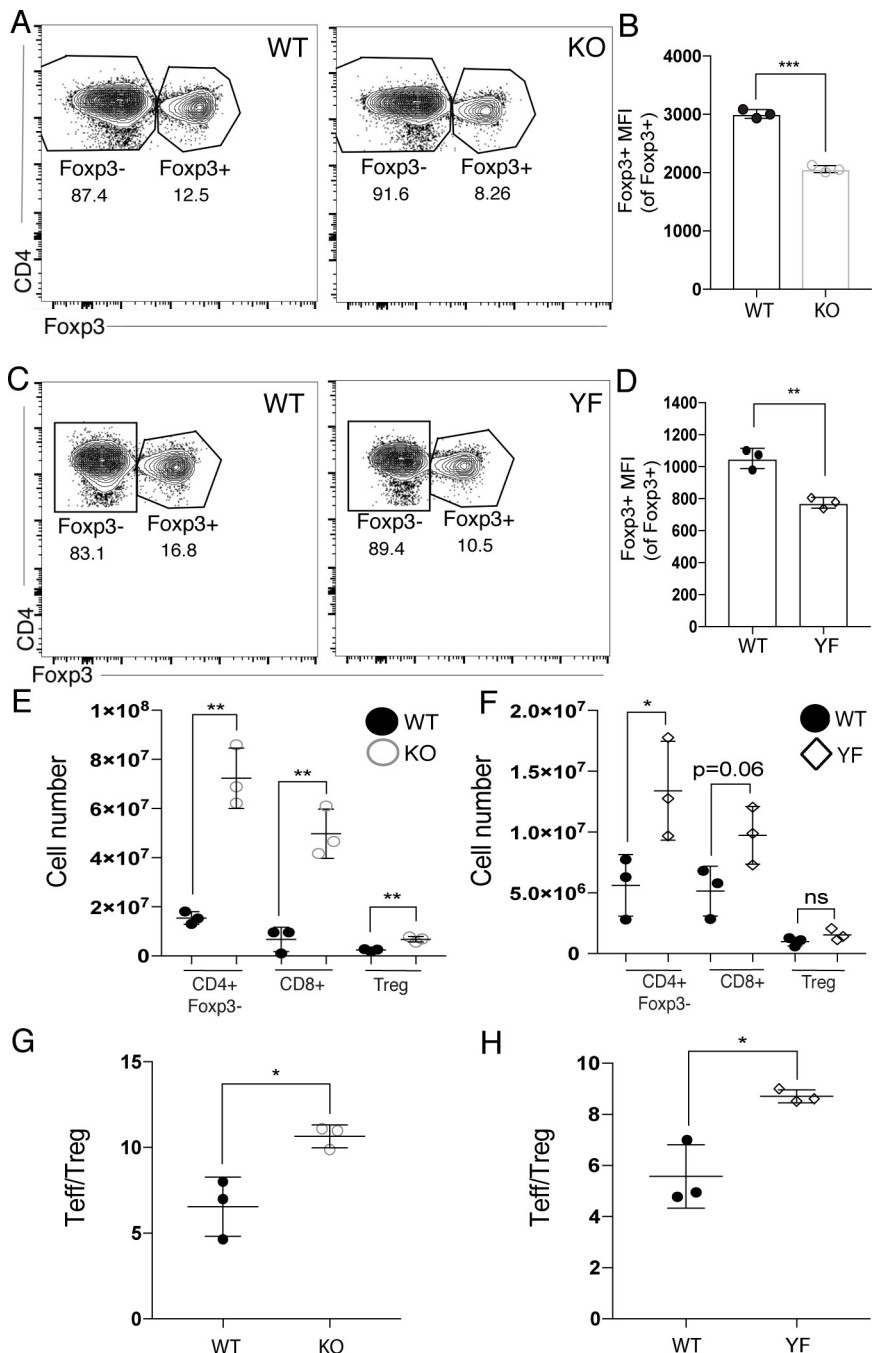

**Fig 4. ICOS KO and ICOS YF mice have baseline defects in Foxp3 expression and Treg frequency, as well as expanded effector T cell populations in the spleen.** **(A)** Representative flow plots showing the decrease in the frequency of Tregs in the spleens of ICOS KO compared to WT controls. **(B)** MFI of Foxp3 in Tregs in the spleen of uninfected WT and ICOS KO mice (n = 3 per group, data is representative of two independent experiments and analyzed by Student's t-test). **(C)** Representative flow plots showing the decrease in the frequency of Tregs in the spleens of ICOS YF mice compared to WT controls. **(D)** MFI of Foxp3 in Tregs in the spleen of uninfected WT and ICOS YF mice (n = 3 per group, data is representative of two independent experiments and analyzed by Student's t-test). **(E-F)** Immune cells were isolated from the spleens of ICOS KO **(E)** and ICOS YF **(F)** and T cell numbers were compared to WT controls (n = 3 per group, data is representative of two independent experiments and analyzed by Student's t-test). The Teff/Treg ratio in the spleen of uninfected mice was calculated in ICOS KO **(G)** and ICOS YF **(H)** mice (n = 3 per group, data is representative of two independent experiments and analyzed by Student's t-test).

was also already present at baseline in ICOS KO and ICOS YF mice, further suggesting a loss of regulation of the effector T cell populations in the spleen during homeostasis. Overall, these data suggest that ICOS KO and ICOS YF mice have similar defects in maintaining normal Treg frequencies and normal Foxp3 expression at baseline, though ICOS KO Tregs are able to maintain Foxp3 and CD25 expression in the inflamed brain during chronic infection, while ICOS YF Tregs are not. The differential effect on the expression of canonical Treg markers in ICOS KO and ICOS YF mice suggests that PI3K-independent signaling downstream of ICOS could play an essential role in maintaining Treg identity particularly during chronic inflammation.

## Discussion

Infection with the parasite *Toxoplasma gondii* leads to a chronic infection in immune-privileged sites such as the central nervous system [45–47]. As such, this chronic infection requires an exquisitely adapted inflammatory response that is capable of controlling the pathogen while also limiting immune-mediated pathology in the brain [48, 49]. Though many aspects of the inflammatory response to chronic *T. gondii* infection have been identified, here we describe a previously unknown multifaceted role for the costimulatory molecule ICOS (inducible T cell costimulator) in the context of chronic infection. Using two models of genetic deficiency in ICOS signaling, constitutive deficiency in ICOS expression (ICOS KO) and constitutive deficiency in ICOS-mediated PI3K signaling (ICOS YF), we found that both genotypes presented similar phenotypes in many aspects of the inflammatory response to *T. gondii* infection. We found that both ICOS KO and ICOS YF mice had similar defects in parasite-specific antibody production and parasite control in the brain, as well as defects in the maintenance of Treg frequency in both the spleen and brain during chronic infection. These results suggest that ICOS signaling plays many roles during the immune response to *T. gondii*, both promoting inflammation and parasite control, while also likely serving to limit inflammation by restricting effector T cell expansion and supporting the Treg population in both the inflamed brain and spleen.

One intriguing difference and paradoxical result observed in ICOS KO versus ICOS YF mice was found in the CNS Treg population. When directly compared during chronic infection, ICOS KO Tregs were able to maintain WT levels of Foxp3 and CD25, while ICOS YF Tregs in the inflamed brain were unable to sustain WT levels of both of these Treg markers. This difference in the maintenance of Foxp3 and CD25 expression in Tregs that completely lack ICOS expression and those that lack ICOS-mediated PI3K signaling is surprising, as PI3K has largely been reported to be the main downstream signaling molecule activated following ICOS ligation [10, 13, 35, 50]. The differential capacity to maintain Foxp3 and CD25 expression in ICOS YF Tregs compared to total ICOS KO Tregs suggests that different intracellular signaling events could be occurring when there is a complete loss of ICOS signaling versus when there is only a loss of ICOS-mediated PI3K signaling. This hypothesis is supported by one report showing that ICOS KO and ICOS YF effector T cells have a differential capacity to confer graft-versus host disease, as ICOS YF T cells were better able to potentiate TCR-mediated calcium flux than total ICOS KO T cells, thereby conferring worse inflammation and disease [51]. This study suggests that not only do PI3K-independent signaling pathways exist downstream of ICOS ligation, but that these pathways could play important roles in the process of inflammation, though the exact signaling pathways involved remain unknown in this context. More recently, a unique motif called IProx was identified in the cytoplasmic tail of ICOS, which was shown to recruit and activate the serine/threonine-protein kinase TBK1 [52]. Although the role of TBK1 (or other as-yet-unidentified ICOS-mediated downstream

signaling pathways) has not been studied in the Treg population specifically, PI3K-independent signaling components such as these are important to consider when studying the role of ICOS signaling in relation to Treg expression of Foxp3 and CD25. Tregs have been shown to be exquisitely sensitive to levels of PI3K activation, where both too little PI3K activation and too much PI3K activation can negatively impact Treg survival and function through downregulation of Foxp3 expression [53, 54]. These results suggest that a balanced level of PI3K activation helps maintain Treg identity and function, but the complex signaling pathways that maintain this balanced PI3K activation in Treg cells remains to be described. Our results suggest that ICOS may promote this balanced level of PI3K activation, possibly through recruitment and activation of negative regulators of PI3K activation such as TBK1 [52]. Overall, understanding how the integration of all ICOS-mediated signals act in maintaining an effective Treg population remains an important area of study as it could provide potential insight into targets for manipulation in human disease.

Mutations in ICOS (or its ligand, ICOSL) in humans are part of a class of mutations leading to the development of common variable immunodeficiency (CVID) [16, 55, 56]. Patients with this immunodeficiency disorder are typically diagnosed following recurrent bacterial infections and present with severely decreased circulating class-switched antibody [16, 55, 57, 58]. Interestingly, however, despite being classified as an immunodeficiency disease, about 20% of patients also present with symptoms of autoimmunity [59]. Though the pathogenesis of this autoimmunity still remains largely unclear, abnormalities in the T cell compartment including expansion of antigen-experienced effector T cells, increased inflammatory cytokine production, and loss of IL-10-producing Tregs have been reported, which could contribute to dysregulation of immune responses and play a role in the autoimmune complications seen in this disease [60–67]. Further understanding of potential autoimmune manifestations in this disease remains an important open question, and our results suggest that the integration of PI3K-dependent and PI3K-independent ICOS signals in the Treg population could potentially play an important role in maintaining immune homeostasis.

## Acknowledgments

The authors would like to thank all the members of the Harris lab, as well as graduate committee members Dr. Kenneth Tung, Dr. Ulrike Lorenz, Dr. Loren Erickson, and Dr. Young Hahn for their insightful comments and discussion during the preparation of this work. We would also like to thank Marieke K. Jones for her helpful input regarding relevant statistical analysis and coding, as well as Dr. Daniel Campbell for his generosity in sharing both ICOS KO and ICOS $Y^{181}F$ mice.

## Author Contributions

**Conceptualization:** Carleigh A. O'Brien, Tajie H. Harris.

**Data curation:** Carleigh A. O'Brien.

**Formal analysis:** Carleigh A. O'Brien.

**Funding acquisition:** Tajie H. Harris.

**Investigation:** Carleigh A. O'Brien.

**Methodology:** Carleigh A. O'Brien.

**Project administration:** Tajie H. Harris.

**Supervision:** Tajie H. Harris.

**Validation:** Tajie H. Harris.

**Writing – original draft:** Carleigh A. O'Brien.

**Writing – review & editing:** Carleigh A. O'Brien, Tajie H. Harris.

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
