## [Decision Letter · Decision Letter 0]

17 Oct 2019

PONE-D-19-26867

ICOS-deficient and ICOS YF mutant mice fail to control Toxoplasma gondii infection of the brain

PLOS ONE

Dear Dr. Harris,

Thank you for submitting your manuscript to PLOS ONE. After careful consideration, we feel that it has merit but does not fully meet PLOS ONE’s publication criteria as it currently stands. Therefore, we invite you to submit a revised version of the manuscript that addresses the points raised during the review process.

We would appreciate receiving your revised manuscript by 12/13/19. If you require additional time, please let us know. To enhance the reproducibility of your results, we recommend that if applicable you deposit your laboratory protocols in protocols.io, where a protocol can be assigned its own identifier (DOI) such that it can be cited independently in the future. For instructions see: http://journals.plos.org/plosone/s/submission-guidelines#loc-laboratory-protocols

We look forward to receiving your revised manuscript.

Kind regards,

Laura J. Knoll

Academic Editor

PLOS ONE

Journal Requirements:

We acknowledge support from the Research Histology core facility at the University of Virginia.

This work was funded by the National Institutes of Health grants R01NS091067 and R56NS106028 to T.H.H. and T32AI007496 to C.A.O.

Reviewers' comments:

Reviewer's Responses to Questions

**Comments to the Author**

1. Is the manuscript technically sound, and do the data support the conclusions?

Reviewer #1: Partly

Reviewer #2: Yes

Reviewer #3: Yes

2. Has the statistical analysis been performed appropriately and rigorously? 

Reviewer #1: Yes

Reviewer #2: Yes

Reviewer #3: Yes

3. Have the authors made all data underlying the findings in their manuscript fully available?

Reviewer #1: Yes

Reviewer #2: Yes

Reviewer #3: Yes

4. Is the manuscript presented in an intelligible fashion and written in standard English?

Reviewer #1: Yes

Reviewer #2: Yes

Reviewer #3: Yes

5. Review Comments to the Author

Reviewer #1: This manuscript describes the role of ICOS in resistance to chronic Toxoplasma infection. The authors compare responses of total ICOS KO mice with ICOS YF mice which retain ICOS expression but cannot signal through PI-3K. For the most part, the authors find these strains are comparable, with increased numbers of effector T cells, a corresponding decrease in FoxP3 positive Treg during chronic infection, and a decrease in the antibody response. This is predictable, but still informative, based upon the notion PI-3K is the major signaling pathway triggered by ICOS.

The major concern with this paper are results in Figure 2, where the total ICOS KO phenocopies the WT strain whereas the ICOS YF mutant display decreased FoxP3 MFI and decreased CD25 expression on Treg. This is difficult or impossible to understand, since the ICOS KO clearly must also be defective in ICOS-triggered PI-3K signaling. The authors conclude in the Abstract and elsewhere that these findings "suggest PI3K-independent effects of ICOS on Treg phenotype", but this is not in line with their data showing an effect of abolishing PI-3K signaling on Treg phenotype. This discrepancy must be resolved.

Other concerns:

Fig. 1C. Show percent positive Bcl-2 in addition to MFI.

Fig. 2A-D. Show also absolute numbers of FoxP3+ cells in addition to percentages. Since the FoxP3- cell number is expanded in the mutant mice, it is possible that there is no difference on Treg number.

Fig. 3. The dramatic effect of ICOS on antibody responses is striking, but no direct evidence is provided that this results in increased susceptibility show in the figure. The authors could determine whether anti-serum infusion corrected the decrease in resistance.

Reviewer #2: In this report, Taije and colleagues investigated the role of ICOS signalling via PI3K during Toxoplasma gondii infection. This is an extension to their original studies, recently published in the Journal of immunology (O’Brien CA et al., 2019). In the present study, their findings suggest that this signalling pathway may be important in supporting regulatory T cell homeostasis. Their studies are technically sound and may be important for the readership at PLoS One. Some minor changes to the current text and figures may improve the quality of this manuscript for publication purposes.

Figures:

1. The authors should consider adding representative flow cytometry plots of the data presented in data in figure one.

2. Axis labels should be added to FACS plots presented in figure 2A-C.

3. Axis labels should be added to FACS plots presented in figure 4A and 4C.

Text:

• The authors should consider removing figure legends from the main text of the manuscript. These should go to a separate section.

Minor scientific comment

• Did the authors measure inflammatory cytokines in the brains of WT and ICOSKO/YF mice? In addition, did the authors examine the phenotype of the effector cells recruited to the brains during infection. If so, this data may be a good addition to the manuscript.

Reviewer #3: The authors present a study to understand the role of ICOS in response to chronic T. gondii infection. Two mutant mouse models are used, one full knock-out and one with a single amino acid substitution (Y to F) to prevent the recruitment and activation of PI3K. Results found that the overall T-cell numbers in the brain is increased in both mutant mouse lines after T. gondii infection, although the number of Treg cells were reduced. Both mutations caused a lack of an antibody response to T. gondii, which in turn, resulted in a higher brain cyst burden. The major differences between the KO and YF mutants were the reduction of Foxp3 expression on Treg cells and the number of CD25+ Treg cells, both of which were reduced in the YF compared to the full KO.

Major Comments:

1. Clarify why it’s stated (lines 294-295) that Foxp3 defects are only in Treg cells of the brain, when they show they are reduced in brain and spleen (Fig 2E and H).

2. The authors state multiple times in the results (e.g. Lines 256 and 291) that CD25 expression levels are changing, however, that data is not shown. Data in Fig 2 shows the percentage of CD25+ cells but not the intensity of their expression. The data and/or text needs to be changed to match one another.

3. To go along with the cyst data (Fig 3), authors should show if either of the mouse mutations have an effect on the survival or health of the mice up to chronic infection.

4. It’s interesting that lack of PI3K signaling reduced Foxp3 expression and CD25 expressing cells but the KO did not, since this mutation also lacks the PI3K signaling. Although this starts to get explained in the discussion, more clarification as to how lacking both PI3K and the other potential cytoplasmic signaling can revert the phenotype back to wild type, since this is all on one protein can be a little counterintuitive.

Minor Comments:

1. References as to how the mutations were generated need to be included in the methods, or more clearly stated elsewhere in the text.

2. Definitions of BMNC and MFI are never stated.

3. Figure 1A: should have “CD8+” (missing the + symbol in the panel).

4. Figure 2A-C: Label the graph axes.

5. Figure 2D-F: For clarity and consistency, label these as being from the brain.

6. Figure 4: Label axes of the flow graphs.

6. PLOS authors have the option to publish the peer review history of their article (what does this mean?). If published, this will include your full peer review and any attached files.

Reviewer #1: No

Reviewer #2: No

Reviewer #3: No

---

## [Author Response · Author response to Decision Letter 0]

12 Dec 2019

Review Comments to the Author

The major concern with this paper are results in Figure 2, where the total ICOS KO phenocopies the WT strain whereas the ICOS YF mutant display decreased FoxP3 MFI and decreased CD25 expression on Treg. This is difficult or impossible to understand, since the ICOS KO clearly must also be defective in ICOS-triggered PI-3K signaling. The authors conclude in the Abstract and elsewhere that these findings "suggest PI3K-independent effects of ICOS on Treg phenotype", but this is not in line with their data showing an effect of abolishing PI-3K signaling on Treg phenotype. This discrepancy must be resolved.

We agree with the reviewer that this is a surprising result! Nevertheless, it was a robust and reproducible observation. We hypothesize that although both ICOS KO and ICOS YF Tregs lack the ability to activate PI3K, ICOS YF Tregs presumably have other intact signaling pathways downstream of ICOS. We hypothesize that these pathways (which are abolished in ICOS KO Tregs) are what is impacting the ability of ICOS YF Tregs to maintain Foxp3 and CD25 in the context of the inflamed brain. For this reason, we expect these particular results to be of interest to researchers that study ICOS signaling.

Other concerns:

Fig. 1C. Show percent positive Bcl-2 in addition to MFI.

All effector T cells in the brain during chronic infection express some level of Bcl-2, therefore the percent positive for all genotypes is 100%. However, the levels of Bcl-2 expression on a per-cell basis changes in the CD4+ effector T cell compartment in ICOS KO and ICOS YF mice. Representative histograms of Bcl-2 expression are now included in Figure 1G for clarity. 

Fig. 2A-D. Show also absolute numbers of FoxP3+ cells in addition to percentages. Since the FoxP3- cell number is expanded in the mutant mice, it is possible that there is no difference on Treg number.

We agree with the reviewer that absolute number and percentages are relevant regarding Tregs. The requested data are now included in Figure 2E and described in the text in lines 295-6.

Fig. 3. The dramatic effect of ICOS on antibody responses is striking, but no direct evidence is provided that this results in increased susceptibility show in the figure. The authors could determine whether anti-serum infusion corrected the decrease in resistance.

We agree with the reviewer that we have not directly implicated antibody defects in the control of cyst burden. For this manuscript, we are reporting the similarities and differences between ICOS KO and ICOS YF mice during infection with T. gondii. We do not feel that we overstated our results and contend that it is merely our hypothesis that antibody defects are resulting in increased cyst burdens. 

Reviewer #2: In this report, Taije and colleagues investigated the role of ICOS signalling via PI3K during Toxoplasma gondii infection. This is an extension to their original studies, recently published in the Journal of immunology (O’Brien CA et al., 2019). In the present study, their findings suggest that this signalling pathway may be important in supporting regulatory T cell homeostasis. Their studies are technically sound and may be important for the readership at PLoS One. Some minor changes to the current text and figures may improve the quality of this manuscript for publication purposes.

Figures:

1. The authors should consider adding representative flow cytometry plots of the data presented in data in figure one.

We agree that the flow plots add more information for the reader. These data are now included in Figure 1C, Figure 1E, and Figure 1G.

2. Axis labels should be added to FACS plots presented in figure 2A-C.

Axis labels have been added to these plots for increased clarity.

3. Axis labels should be added to FACS plots presented in figure 4A and 4C.

Axis labels have been added to these plots for increased clarity.

Text:

The authors should consider removing figure legends from the main text of the manuscript. These should go to a separate section.

 The figure legends have now been embedded in the Results section as per PLoS One guidelines. 

Minor scientific comment

Did the authors measure inflammatory cytokines in the brains of WT and ICOSKO/YF mice? In addition, did the authors examine the phenotype of the effector cells recruited to the brains during infection. If so, this data may be a good addition to the manuscript.

We agree that an analysis of the effector T cells is also of great interest. Because we observed changes in CD25 and Bcl-2 expression in our 2019 publication, we examined the number of effector T cells in the brain along with their expression of CD25 and Bcl-2. We have not examined cytokine production, but expect results to be similar to anti-ICOSL treatment, where IFN-� production does not change on a per cell basis, but there is an increase in effector cells in the brain leading to an overall increase in cytokine production. 

Reviewer #3: The authors present a study to understand the role of ICOS in response to chronic T. gondii infection. Two mutant mouse models are used, one full knock-out and one with a single amino acid substitution (Y to F) to prevent the recruitment and activation of PI3K. Results found that the overall T-cell numbers in the brain is increased in both mutant mouse lines after T. gondii infection, although the number of Treg cells were reduced. Both mutations caused a lack of an antibody response to T. gondii, which in turn, resulted in a higher brain cyst burden. The major differences between the KO and YF mutants were the reduction of Foxp3 expression on Treg cells and the number of CD25+ Treg cells, both of which were reduced in the YF compared to the full KO.

Major Comments:

1. Clarify why it’s stated (lines 294-295) that Foxp3 defects are only in Treg cells of the brain, when they show they are reduced in brain and spleen (Fig 2E and H).

We thank the reviewer for pointing out the lack of clarity in the text regarding these results. The text describing these results has now been changed to describe in more detail the Treg defects in the brain and spleen in lines 316-329. 

2. The authors state multiple times in the results (e.g. Lines 256 and 291) that CD25 expression levels are changing, however, that data is not shown. Data in Fig 2 shows the percentage of CD25+ cells but not the intensity of their expression. The data and/or text needs to be changed to match one another.

The data showing the MFI of CD25+ Tregs in both the brain and spleen are now included in Figure 2H and Figure 2L, respectively. These results are described in the text in lines 316-325. 

3. To go along with the cyst data (Fig 3), authors should show if either of the mouse mutations have an effect on the survival or health of the mice up to chronic infection.

Though both ICOS KO and ICOS YF mice survived the acute stage of infection and the early chronic phase in our hands, we did observe increased sickness in these mice compared to WT (increased ruffled fur, less activity), suggesting that the inability to control parasite burden in the brain might lead to survival defects in later chronic stages of infection.

4. It’s interesting that lack of PI3K signaling reduced Foxp3 expression and CD25 expressing cells but the KO did not, since this mutation also lacks the PI3K signaling. Although this starts to get explained in the discussion, more clarification as to how lacking both PI3K and the other potential cytoplasmic signaling can revert the phenotype back to wild type, since this is all on one protein can be a little counterintuitive.

It has been suggested in the literature that Tregs are very sensitive to PI3K activation levels, where too much or too little PI3K activation can negatively impact their survival and function (Abu-Eid, et al., 2014 & Sauer, et al. 2008). It is possible that the additional signaling pathways downstream of ICOS besides PI3K (which are intact in ICOS YF Tregs) serve as a pathway to negatively regulate activation of PI3K, as has been shown in Tfh cells (Pedros, et al. 2016). If this is true in Tregs, WT Tregs might recruit and activate PI3K downstream of ICOS, but also activate these negative regulatory pathways that prevent “too much” PI3K activation, leading to balanced PI3K activation in the Treg. In ICOS YF Tregs, the inability to activate PI3K is lost, so only these potential negative pathways are activated downstream of ICOS. If only these pathways are activated downstream of ICOS in ICOS YF Tregs, they might negatively impact PI3K too much, leading to too little PI3K activation, with has also been shown to decrease Treg Foxp3 expression. In ICOS KO Tregs, both the positive PI3K signal and the negative signal are lost, meaning the “balance” is maintained enough (possibly through other PI3K activating signals that can compensate for the loss of ICOS such as TCR/MHC, CD28, or IL-2) for ICOS KO Tregs to maintain Foxp3 and CD25 expression in the inflamed brain. Additional comments regarding this hypothesis have been added to the discussion in lines 464-472. 

Minor Comments:

1. References as to how the mutations were generated need to be included in the methods, or more clearly stated elsewhere in the text.

References to the original papers describing the generation of these mice have been added to the Methods section (line 109).

2. Definitions of BMNC and MFI are never stated.

These terms have been defined in the Figure Legend for Figure 1 (lines 213 and 216-7, respectively).

3. Figure 1A: should have “CD8+” (missing the + symbol in the panel).

This has been added to Figure 1A.

4. Figure 2A-C: Label the graph axes.

Axis labels have been added to flow plots for clarity.

5. Figure 2D-F: For clarity and consistency, label these as being from the brain.

The graphs in in Figure 2D-F have now been labeled as showing data from the brain.

6. Figure 4: Label axes of the flow graph

Axis labels have been added to these flow plots for increased clarity.

---

## [Decision Letter · Decision Letter 1]

13 Jan 2020

ICOS-deficient and ICOS YF mutant mice fail to control Toxoplasma gondii infection of the brain

PONE-D-19-26867R1

Dear Dr. Harris,

We are pleased to inform you that your manuscript has been judged scientifically suitable for publication and will be formally accepted for publication once it complies with all outstanding technical requirements. We also ask that you add a comment about the seemingly paradoxical result of a Treg phenotype in ICOS YF mice, but no phenotype in KO cells, as suggested by reviewers 1 and 3. Also please add axis labels to figure 4 A and C. 

With kind regards,

Laura J. Knoll

Academic Editor

PLOS ONE

Additional Editor Comments (optional):

Reviewers' comments:

Reviewer's Responses to Questions

**Comments to the Author**

1. If the authors have adequately addressed your comments raised in a previous round of review and you feel that this manuscript is now acceptable for publication, you may indicate that here to bypass the “Comments to the Author” section, enter your conflict of interest statement in the “Confidential to Editor” section, and submit your "Accept" recommendation.

Reviewer #1: (No Response)

Reviewer #2: All comments have been addressed

Reviewer #3: (No Response)

2. Is the manuscript technically sound, and do the data support the conclusions?

Reviewer #1: Yes

Reviewer #2: Yes

Reviewer #3: Yes

3. Has the statistical analysis been performed appropriately and rigorously? 

Reviewer #1: Yes

Reviewer #2: N/A

Reviewer #3: Yes

4. Have the authors made all data underlying the findings in their manuscript fully available?

Reviewer #1: Yes

Reviewer #2: Yes

Reviewer #3: Yes

5. Is the manuscript presented in an intelligible fashion and written in standard English?

Reviewer #1: Yes

Reviewer #2: Yes

Reviewer #3: Yes

6. Review Comments to the Author

Reviewer #1: The authors have addressed most of the reviewers' comments satisfactorily. My only issue remains with the seemingly paradoxical result that the authors report no Treg phenotype with ICOS KO but a phenotype with the ICOS YF mice. The authors provide a nice explanation of why this might be the case in the response to reviewer 3, but it is still glossed over in the manuscript. I suggest that the authors incorporate their response to the reviewer in the manuscript discussion.

Reviewer #2: I am satisfied with authors' responses to my comments. Therefore, I have no further concerns on this submission.

Reviewer #3: The authors did a great job addressing many of the review concerns, though I have two comments I would like addressed further.

1. I would like to see the author comments addressing Reviewer 3 Major Comment 3 added to the text of the manuscript (I believe even stating the phenotype with “data not shown” would be sufficient). I believe this information would be beneficial to the Toxoplasma community since the mice survive infection while lacking antibodies. Also, it is important to know that many of the mice survived so that we know that the data is not skewed by looking at only the rare mouse that happened to live to the chronic stage.

2. Panels Figure 4 A and C do not have axis labels. Please label for clarification.

7. PLOS authors have the option to publish the peer review history of their article (what does this mean?). If published, this will include your full peer review and any attached files.

Reviewer #1: No

Reviewer #2: No

Reviewer #3: No

---

## [Editor Report · Acceptance letter]

16 Jan 2020

PONE-D-19-26867R1 

ICOS-deficient and ICOS YF mutant mice fail to control *Toxoplasma gondii* infection of the brain 

Dear Dr. Harris:

I am pleased to inform you that your manuscript has been deemed suitable for publication in PLOS ONE. Congratulations! Your manuscript is now with our production department. 

With kind regards,

on behalf of

Prof. Laura J. Knoll 

Academic Editor

PLOS ONE